# Neural drive and motor unit characteristics after anterior cruciate ligament reconstruction: implications for quadriceps weakness

David A. Sherman[1,2,3], Justin Rush[4], Matt S. Stock[5], Christopher D. Ingersoll[6] and Grant E. Norte[5]

[1] Live4 Physical Therapy and Wellness, Acton, Massachusetts, United States of America
[2] Chobanian & Avedisian School of Medicine, Boston University, Boston, Massachusetts, United States of America
[3] Harvard John A. Paulson School of Engineering and Applied Sciences, Harvard University, Cambridge, Massachusetts, United States of America
[4] Division of Physical Therapy, School of Rehabilitation and Communication Sciences, College of Health Sciences and Professions, Ohio University, Athens, Ohio, United States of America
[5] Cognition, Neuroplasticity, & Sarcopenia (CNS) Lab, Institute of Exercise Physiology and Rehabilitation Science, University of Central Florida, Orlando, FL, United States of America
[6] College of Health Professions and Sciences, School of Kinesiology and Rehabilitation Sciences, University of Central Florida, Orlando, Florida, United States of America

Corresponding author
David A. Sherman, dsherm@bu.edu

## ABSTRACT

**Purpose:** The purpose of this investigation was to compare the quality of neural drive and recruited quadriceps motor units' (MU) action potential amplitude ($MUAP_{AMP}$) and discharge rate (mean firing rate (MFR)) relative to recruitment threshold (RT) between individuals with anterior cruciate ligament reconstruction (ACLR) and controls.

**Methods:** Fourteen individuals with ACLR and 13 matched controls performed trapezoidal knee extensor contractions at 30%, 50%, 70%, and 100% of their maximal voluntary isometric contraction (MVIC). Decomposition electromyography (dEMG) and torque were recorded concurrently. The Hoffmann reflex (H-reflex) and central activation ratio (CAR) were acquired bilaterally to detail the proportion of MU pool available and volitionally activated. We examined $MUAP_{AMP}$-RT and MFR-RT relationships with linear regression and extracted the regression line slope, y-intercept, and RT range for each contraction. Linear mixed effect modelling used to analyze the effect of group and limb on regression line slope and RT range.

**Results:** Individuals with ACLR demonstrated lower MVIC torque in the involved limb compared to uninvolved limb. There were no differences in H-reflex or CAR between groups or limbs. The ACLR involved limb demonstrated smaller mass-normalized RT range and slower MU firing rates at high contraction intensities (70% and 100% MVIC) compared to uninvolved and control limbs. The ACLR involved limb also demonstrated larger MU action potentials in the VM compared to the contralateral limb. These differences were largely attenuated with relative RT normalization.

**Conclusions:** These results suggest that persistent strength deficits following ACLR may be attributable to a diminished quadriceps motor neuron pool and inability to upregulate the firing rate of recruited MUs.

## INTRODUCTION

Anterior cruciate ligament (ACL) injury is common, with an incidence of 250,000 injuries in the United States annually (*Mather et al., 2013*). Individuals who undergo gold-standard ACL reconstruction (ACLR) treatment self-report poor function (*Norte et al., 2018a*) and experience high rates of re-injury (*Otzel, Chow & Tillman, 2015*) and osteoarthritis (*Norte et al., 2018a*; *Øiestad et al., 2022*). Persistent quadriceps weakness remains the hallmark clinical impairment following ACLR and is highly predictive of both poor self-reported function and irreversible osteoarthritis (*Norte et al., 2018a*; *Øiestad et al., 2022*). Current clinical practice strategies fail to restore adequate quadriceps function despite targeting strength training after ACLR (*Lisee et al., 2019*).

Muscle function depends on the quality of neural drive to the muscle. The resultant muscle output is determined in part by the recruited motor units' (MU) properties. Specifically, force output is highly dependent on the summation of the size, firing rates, and recruitment thresholds of the recruited MU within the muscle. Following ACL injury, arthrogenic muscle inhibition limits quadriceps muscle function by causing reflexive inhibition of the quadriceps motor neuron pool. Arthrogenic muscle inhibition manifests clinically as profound weakness, voluntary activation failure, and atrophy of the uninjured quadriceps muscle (*Lisee et al., 2019*; *Pietrosimone et al., 2022*). While a natural response to protect an injured region acutely following injury, the inability to recruit MU limits responsiveness to strength training in rehabilitation. As such, failing to overcome arthrogenic muscle inhibition presents a limiting factor in recovery from joint injury (*Hopkins & Ingersoll, 2000*) and may explain persistent muscle weakness and atrophy after ACL injury.

Fewer available MU, due to arthrogenic muscle inhibition, may limit neural recruitment strategies to perform quadriceps strengthening exercises (*Lepley et al., 2015*; *Pietrosimone et al., 2022*). Fewer available MU theoretically necessitates an upregulation of the firing rate to preserve force generation (*Maffiuletti et al., 2016*). As quadriceps motor neuron pool excitability normalizes (*Lepley et al., 2015*), there remains a persistent inability to voluntarily recruit MU (*i.e.*, quadriceps central activation failure) (*Lisee et al., 2019*; *Otzel, Chow & Tillman, 2015*). In light of this evidence, some theorize (*Lepley et al., 2020*; *Noehren et al., 2016*; *Rowan et al., 2012*) that individuals habituate a compensated neuromuscular strategy, primarily recruiting only what MU were available early in recovery, while inhibited MU are not recruited and thus catabolized over time. Due to limitations in technology, traditional approaches used to study this phenomenon fail to fully describe MU properties after ACLR, which limits the ability to design and prescribe effective treatments.

Traditional methods in rehabilitative research include gross assessment of summated MU action potentials using surface electromyography (EMG), quadriceps motor neuron pool excitability using the Hoffmann reflex (H-reflex), or the proportion of voluntary

quadriceps activation (central activation ratio, CAR) using the superimposed burst technique. Although informative, none of these approaches account for the individual MU properties: MU action potential amplitude (MUAP$_{AMP}$) and discharge rate (mean firing rate (MFR)) relative to recruitment threshold (RT) (*Maffiuletti et al., 2016*). Recently, the ability to study MU properties using EMG signal decomposition has been described (*Nawab, Chang & De Luca, 2010*) and independently validated (*De Luca, Nawab & Kline, 2015*; *Martinez-Valdes et al., 2016*), allowing for the study of specific adaptations within the motor neuron pool (*Sterczala et al., 2020*).

In healthy muscle, larger MUs with lower firing rates are recruited as volitional torque output increases creating negative (MUAP$_{AMP}$-RT) and positive (MFR-RT) linear relationships respectively (*De Luca & Contessa, 2015*). Decomposition of EMG signal has revealed evidence of changes in the slopes of these relationships in response to cryotherapy (*Mallette et al., 2018*), stroke (*Hu et al., 2016*), disuse/immobilization (*MacLennan et al., 2021*), and periods of strength and conditioning (*Pope et al., 2016*; *Sterczala et al., 2020*; *Stock & Thompson, 2014*). For example, 8-weeks of hypertrophy-based strength training results in recruitment of larger MUs at lower relative torque output (*e.g.*, steeper MUAP$_{AMP}$-RT slope) (*Pope et al., 2016*; *Sterczala et al., 2020*) without change in MFR (*Stock & Thompson, 2014*), suggesting that increases in strength and muscle cross-sectional area are highly associated with MU size (*Pope et al., 2016*; *Sterczala et al., 2020*). Whereas neural inhibition after stroke disrupts the relationship between MU firing rate and torque, suggesting a less optimal schema for recruitment of fast MUs at low volitional effort and slow MUs at high volitional effort (*Hu et al., 2016*). A recent study (*Nuccio et al., 2021*) suggests that individuals with ACLR may exhibit a similar phenomenon, as they may be less able to upregulate MU discharge rates in the ACLR limb compared to uninvolved limb. Such impairments are thought to stem from neural inhibition (*Hu et al., 2016*; *Nuccio et al., 2021*).

Considering the extent of quadriceps weakness, activation failure, and atrophy after ACLR, recruited MUs may also be smaller at any given RT (*e.g.*, more gradual MUAP$_{AMP}$-RT slope), suggestive of deconditioning or MU atrophy. However, if the slopes of these relationships are similar in the ACLR limbs, quadriceps weakness may suggest a general inability to recruit MUs during high intensity contractions (*i.e.*, activation failure). This may be measured as the RT range, which represents the length of recruitment curve between the first and last recorded MUs. In this way, a better understanding of the relationships between MUs size, firing rate, and RT with respect to quadriceps motor neuron pool excitability and voluntary activation following ACLR will inform physical rehabilitation of muscle function.

Therefore, the purpose of this investigation was to compare relationships of quadriceps MU properties between individuals with ACLR and controls. We analyzed the relationships between MU action potential size, firing rate, and RT in the vastus medialis (VM) and vastus lateralis (VL) during submaximal and maximal isometric knee extension trials in the involved, uninvolved, and matched limbs in both groups. As a comprehensive assessment of neuromuscular function, we measured quadriceps motor neuron pool excitability, voluntary activation, and strength to fully appreciate the context of the

observed relationships between MU properties. On the basis of prior work, we anticipated that quadriceps strength and voluntary activation would be lower in the involved limb (*Lisee et al., 2019*), yet motor neuron pool excitability would be normal (*Rush, Glaviano & Norte, 2021*). We hypothesized that individuals with ACLR would demonstrate faster MU firing rates (*e.g.*, more gradual MFR-RT slope) and smaller MU action potential size (*e.g.*, more gradual MUAP$_{AMP}$-RT slope) with increasing volitional torque output. We also hypothesized that the RT range would be shorter in the involved compared to uninvolved and control limbs.

## MATERIALS AND METHODS

We used a cross-sectional design to assess the relationships between quadriceps MU properties of bilateral lower extremities in individuals with ACLR and matched controls. Limb and group were treated as independent variables (*i.e.*, involved, uninvolved, matched limbs). The slope the relationships between MFR-RT, and MUAP$_{AMP}$-RT from the VM and VL, as well as the RT range, were treated as dependent variables. Measures of H-reflex, mass-normalized maximum voluntary isometric contraction (MVIC) torque, CAR, average torque generated during contractions, and degree of error from target force are reported descriptively. The y-intercepts of each relationship were treated as *a priori* defined co-variates given their potential influence on the outcomes of interest. Outcomes were assessed separately during four contraction intensities (30%, 50%, 70%, and 100% MVIC). We assessed dependent variables while normalizing RT to participant's body mass (Nm/kg), as well as relative to maximal volitional strength (%MVIC), as each offers distinct implications. Namely, it is important to note that MVIC normalization can be misleading as it diminishes in the magnitude of expected differences in strength between groups and limbs, leading to an artificially elongated RT ranges and slopes. On the other hand, mass-normalized RT considers the individual's ability to generate torque relative to their body mass, is the gold standard for assessment of strength recovery following ACLR and may provide a more comprehensive understanding of the underlying MU properties related to clinical muscle weakness after ACLR.

Individuals with a history of primary unilateral ACLR and sex-, age-, and activity-level matched controls volunteered to participate in this study. The ACLR group must have been 18–35 years old and greater than 6 months from surgery to be eligible. Those with history of failed reconstruction, prior multi-ligament knee injury, or knee osteoarthritis were excluded. Graft type and time from surgery were reported descriptively. The control group was without history of injuries or surgeries to either lower extremity. Any person with known history of concussion in the past 6 months, neurological disorder, or who were taking medications which may alter neural excitability (*e.g.*, stimulants and depressants) were excluded. Participants were instructed to avoid caffeine and alcohol for 24 h prior to testing (*Sherman et al., 2023*). The University of Toledo Institutional Review Board for Biomedical Research approved this study (IRB # 300052-UT). All participants provided verbal and written informed consent prior to beginning the study procedures.

All participants self-reported their knee function (International Knee Documentation Committee (IKDC) Subjective Knee Evaluation), activity level (Tegner Activity Scale and

International Physical Activity Questionnaire (IPAQ)), and psychological readiness for sport activities (Anterior Cruciate Ligament Return to Sport after Injury (ACL-RSI) scale). Testing procedures were conducted during one laboratory session. The order of limb testing was randomized. Control limbs were matched to ACLR limbs by leg dominance (*i.e.*, left involved limb of right-limb-dominant participant matched to non-dominant limb of control).

The H-reflex was used to quantify motor neuron pool excitability of the VL. Participants lay supine with knees flexed to approximately 15° using a bolster and with hands resting on lap. Two 10-mm pre-gelled Ag-AgCl recording electrodes (EL 503; BIOPAC Systems, Inc., Goleta, CA, USA) with an interelectrode distance of 20 mm were placed on cleaned, shaved, debrided skin according to SENIAM guidelines (*Hermens et al., 2000*). A bipolar stimulation electrode was positioned in the inguinal fold over the femoral nerve. Participants wore earplugs, closed their eyes, and were instructed to "clear the mind" throughout testing. A series of 1-millisecond square-wave electrical stimuli ranging from 10 to 200 V were delivered *via* a stimulator module (model STM100A; BIOPAC Systems, Inc., Goleta, CA, USA) and a current isolation unit (model STMISOC; BIOPAC Systems, Inc., Goleta, CA, USA) with a minimum of 10 s between stimuli (*Norte et al., 2018b*). We identified the stimulation intensities that maximized each of H- and M-waves and used a three-trial average amplitude at each intensity to calculate the H:M ratio. Signals were sampled at 2,000 Hz, and band-pass filtered from 10–500 Hz using AcqKnowledge 5.0 software (BIOPAC Systems, Inc., Goleta, CA, USA).

Knee extension MVIC torque and quadriceps CAR were used to quantify strength and voluntary activation. Participants were seated upright in a stationary dynamometer (System 4 Pro; Biodex Inc., Shirley, NY, USA), with the hips flexed to 85 degrees and knees flexed to 90 degrees. In preparation for CAR assessment, two 3″ × 5″ self-adhesive electrodes (ValuTrode, Axelgaard Manufacturing CO., Ltd, Fallbrook, CA, USA) were placed over the VM and proximal VL. Participants were secured using chest and lap straps to reduce aberrant motion during testing. They completed a standardized, progressive, familiarization protocol consisting of progressive increases in perceived effort (25%, 50%, 75% effort) (*Sherman et al., 2023*). During three MVIC trials, we instructed participants to kick out as hard and as fast as possible for five seconds as a TV monitor provided synchronized visual feedback of performance and 110% goal setting. We provided maximal verbal encouragement throughout each trial. We used the average of the trials to determine the torque level for the subsequent CAR stimulation threshold, and submaximal contractions. All torque data were sampled at 2,000 Hz, low-pass filtered at 15 Hz, and normalized to body mass.

The superimposed burst technique was used to estimate quadriceps CAR during two additional MVIC trials. A supramaximal percutaneous electrical stimulus was manually delivered to the quadriceps using a square-wave stimulator (model S88; GRASS-TeleFactor, West Warwick, RI, USA) and isolation unit (model SIU8T; GRASS-TeleFactor, West Warwick, RI, USA). We delivered the stimulus (10-pulse train, 100 Hz, 600 ms pulse duration, 150 V) based on the plateau of the torque output at ≥95% of MVIC threshold (*Garcia et al., 2022*). The induced torque increase was used to calculate the CAR. A

minimum of 60 s rest between test trials was ensured. Testing was then repeated on the opposite limb. Following MVIC and CAR testing, participants were given a minimum of 10 min rest where they were permitted to move around as desired.

Following the 10-min break in procedures, participants returned to the dynamometer for the primary arm of testing. We cleaned and debrided the skin and placed 4-pin surface sensor arrays (Trigno Galileo Sensor, Delsys Inc., Natick, MA, USA) over the VL and VM according to SENIAM guidelines (*Hermens et al., 2000*). Throughout all contractions, EMG signals from each pin in the array were differentially amplified, bandpass filtered from 20–450 Hz, sampled at 2,222 Hz, and stored for subsequent decomposition and analysis. Torque was captured synchronously. A familiarization protocol consisted of practice trials at 20% MVIC and visualization of contraction targets until the participant was comfortable with the task. The testing protocol consisted of both submaximal (30%, 50%, and 70% MVIC) and maximal (100% MVIC) contractions, performed in a random order. The submaximal contractions followed a trapezoid pattern, characterized by a linear increase of torque at a rate of 10% MVIC/s, a plateaued region (15 s for 30% and 50% trials, 10 s for 70% trials), and a 10% MVIC/s linear decrease of torque (*Stock & Mota, 2017*; *Stock & Thompson, 2014*). The maximal contractions followed a similar trapezoid pattern but with a 20% MVIC/s linear increase, a 5 s plateaued region, and a 20% MVIC/s linear decrease of torque. Participants used real-time torque feedback overlaid with a visual template of the isometric trapezoidal shape (*Sherman et al., 2023*). Participants performed three repetitions at each intensity prior to continuing to the next in randomized order. Once again, a minimum of 60 s rest between test trials was ensured. Participants were encouraged to request additional rest as needed.

We assessed fidelity to target using normalized root mean square error (NRMSE) and mean torque output during the plateaued region of each trial. The NRMSE represented the percent of error from the target force, whereas the mean torque output represents the absolute torque production resulting from MU behavior.

The four filtered EMG signals were decomposed into their constituent motor unit action potential trains using NeuroMap Software (v1.2.2; Delsys Inc., Natick, MA, USA) applying the Precision Decomposition III algorithm with accuracy tested *via* the decompose-synthesize-decompose-compare test (*Nawab, Chang & De Luca, 2010*). Only MU that demonstrated ≥90% accuracy were included in subsequent analysis. Visual inspection and editing of spike trains derived from MU decomposition to remove MU of lower quality was then performed (*Del Vecchio et al., 2020*). The resulting MU RT were defined at both mass-normalized RT (Nm/kg) and relative (%MVIC) force levels. The MFR-RT and $MUAP_{AMP}$-RT relationships were quantified using RT during the ascending portion, and $MUAP_{AMP}$ (μV) and MFR (pps) of the plateaued region of the trapezoidal contraction. $MUAP_{AMP}$ was quantified as the average peak-to-peak amplitude of each of the 4-action potential waveforms as previously described (*Hu, Rymer & Suresh, 2013*). Figure 1A depicts representative contractions and relevant feature extraction.

MFR-RT and $MUAP_{AMP}$-RT relationships were analyzed for each participant individually. We examined the relationships with linear regression of MFR and $MUAP_{AMP}$ against RT and extracted the regression line slope and y-intercept for each individual

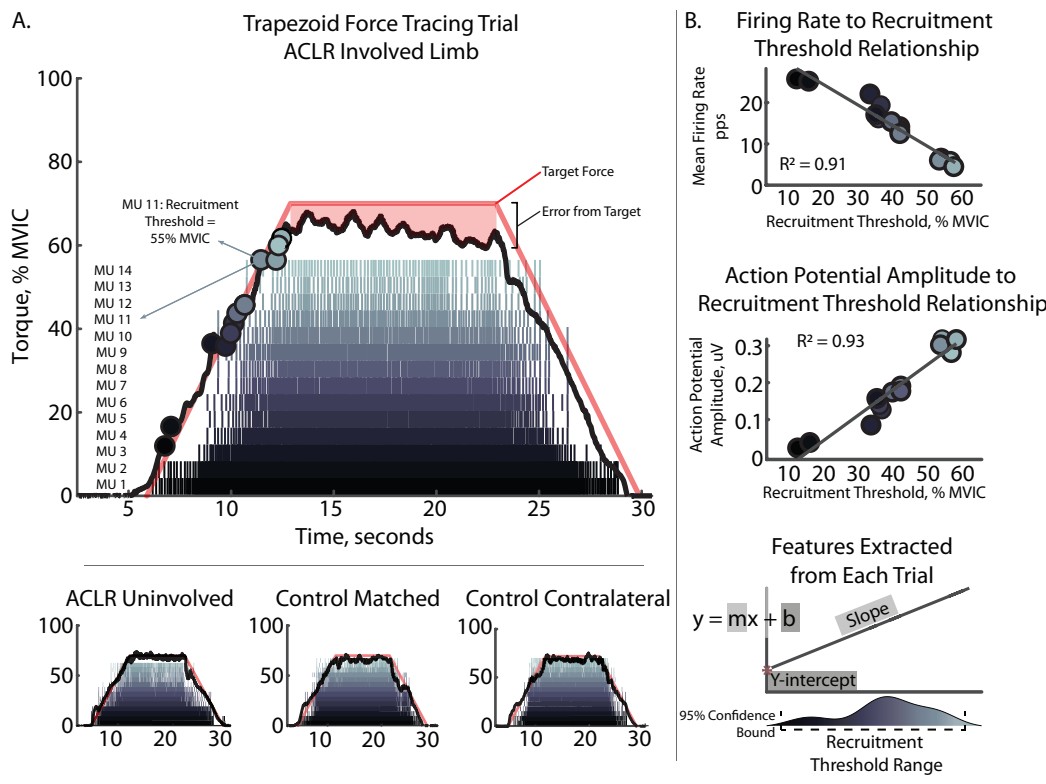

**Figure 1 Characteristic trapezoidal contraction and MU decomposition.** (A) Depicts characteristic trapezoidal contraction and MU decomposition at 70% intensity. Raster plot shows each individual MU Firing with MU number displayed at the left. Solid black line depicts torque output relative to peak torque output during MVIC trial (% MVIC). Colored circles on torque output represent recruitment threshold for corresponding MU. Solid red line depicts visual feedback of target shape with 7 s ramp and 10 s plateau periods. Mean firing rate were taken from the plateaued region. Mean torque and normalized root mean square error were quantified as average torque and the degree of error from the target line during the plateau. Subplots contains additional examples for other limb and groups comparators. (B) Depicts participant specific regressions and feature extraction of recruitment threshold range, slopes, and y-intercepts for the mean firing rate to recruitment threshold and action potential amplitude to recruitment threshold relationships. ACLR, anterior cruciate ligament reconstruction; MVIC, maximum voluntary isometric contraction; MU, motor unit; pps, pulses per second; uV, microvolts.

contraction for use in statistical analysis (Fig. 1B). The MFR-RT slope represents the rate of change in MFR of MU as a function of increasing recruitment force. The $\text{MUAP}_{\text{AMP}}$-RT slope represents the rate of change in $\text{MUAP}_{\text{AMP}}$ of MU as a function of increasing recruitment force. Recruitment threshold range was determined using the 95% confidence bounds of MU RTs for each trial and represents the recruitment curve of the MU from first to last recruited. All regression analyses were performed using a custom MATLAB script (MathWorks, Natick, MA, USA).

Distribution normality was checked through the Shapiro–Wilk test. Outliers were removed if greater than 3 median absolutes deviations from the median. Trials with poorly fit relationships ($R^2 < 0.6$) were also excluded. We utilized a mixed-effects model to analyze our data, taking into account the nested structure of repeated measures within each participant. This approach was chosen to account for the correlation among repeated

measurements and the potential heterogeneity across participants. Repetitions were included as a random factor in the model, allowing for inclusion of all trials and all identified MUs, while accounting for individual-specific variations. By incorporating random effects, we were able to capture both within-subject variability and between-subject differences, improving the robustness of our analysis and providing more accurate estimates of fixed effects. This modeling strategy is recommended for neuromechanical data, as it ensures that the observed effects are not driven by individual differences or measurement variability (*Wilkinson, Mazzo & Feeney, 2023*). Separate models were used to analyze possible between-group and between-limb differences in slope of the MFR-RT and MUAP$_{AMP}$-RT relationships and RT range for each contraction intensity, while controlling for the number of MUs. For analysis of slopes, y-intercept was also included as a co-variate. In the event of group by limb interactions, a *post hoc* correction was applied to account for multiple comparisons using Tukey's Honestly Significant Difference (HSD). In the event of group main effects, an independent *t*-test was applied. To estimate the magnitude of observed differences, we calculated Cohen's d effect sizes with 95% confidence intervals. Effect sizes were interpreted as small (≥0.2), moderate (≥0.5), or large (≥0.8). We conducted the statistical analysis using JMP Pro 16 (SAS Institute Inc., Cary, NC, USA). All statistical tests were two-sided, and a significance level of α = 0.05 with Benjamini-Hochberg False Discovery Rate (FDR) correction was used to determine statistical significance. Results from the 30% and 50% contraction intensities can be found in Supplement 1.

The funding organization had no role in data collection, analysis, or interpretation, nor do they have the right to approve or disapprove publication of the work.

## RESULTS

Thirty-two individuals completed all study procedures. Five participants (three ACLR, two control) demonstrated ≤4 MU during more than three-fourths of contractions and were excluded from analysis due to poor MU decomposition. This left 14 and 13 individuals in the ACLR and control groups, respectively. Participants were similar in age, height, weight, and activity level. Participants reported high levels of physical activity, with 41% participating in NCAA Division one college athletics (n, ACLR = 6, control = 5) and 85% participating in routine strength and conditioning (n, ACLR = 12, control = 11) at the time of collection. The ACLR group demonstrated significantly lower self-reported knee function (IKDC, d = 0.96 [0.17–1.76]), lower confidence (ACL-RSI, d = 1.42 [0.04, 1.61]), and higher kinesiophobia (TSK-11, d = 0.82 [0.58–2.27]) compared to controls (Table 1).

Motor neuron pool excitability was similar between groups and limbs (Table 2). The involved limb was significantly weaker than the uninvolved limb in the ACLR group (d = 1.00 [0.21–1.78]) and contralateral control limb (d = 0.86 [0.07–1.65]), but not different than control limbs (Table 2). Quadriceps MVIC limb symmetry index was lower in the ACLR group compared to controls (d = 0.96 [0.16–1.75]). Quadriceps CAR and H-reflex were similar between groups and limbs.

The analysis of task fidelity and torque output revealed both ACLR group and ACLR limb impairments (Fig. 2). During the 100% trials, the ACLR group demonstrated higher

**Table 1 Participant demographics.**

|  | ACLR (n = 14) | Control (n = 13) | p value |
|---|---|---|---|
| Age, years | 20.9 ± 2.5 | 21.8 ± 1.8 | 0.680 |
| Sex (% female) | 50.0% | 46.2% |  |
| Height, cm | 177.8 ± 9.2 | 173.2 ± 11.4 | 0.220 |
| Weight, kg | 73.4 ± 8.3 | 74.1 ± 13.1 | 0.873 |
| Tegner activity scale | 7.9 ± 2.0 | 8.0 ± 1.4 | 0.909 |
| IPAQ, MET/minute/week | 2,495.3 ± 1,459.8 | 1,677.4 ± 1,140.5 | 0.354 |
| IKDC | 88.8 ± 8.9 | 96.2 ± 6.1 | **0.019** |
| TSK-11 | 19.4 ± 3.5 | 16.6 ± 3.3 | **0.031** |
| ACL-RSI | 70.0 ± 21.1 | 94.2 ± 11.0 | **0.001** |
| Graft type | 7 BTB (50.0%) 7 HT (50.0%) |  |  |
| Time after surgery, months | 28.7 ± 24.5 |  |  |
|  | Range [7, 79] |  |  |

Note:
SD, standard deviation; ACLR, anterior cruciate ligament reconstruction; IPAQ, international physical activity questionnaire; MET, metabolic equivalent unit; IKDC, international knee documentation committee; TSK-11, tampa scale of kinesiophobia 11-item; ACL-RSI, anterior cruciate ligament readiness for sport index; BTB, bone-tendon-bone; HT, hamstrings tendon. Bold indicates statistically significant differences between groups.

**Table 2 Group and limb comparison of co-variates.**

|  | ACLR | | Control | | $F_{(3,54)}$ | p value |
|---|---|---|---|---|---|---|
|  | Involved | Uninvolved | Matched involved | Matched contralateral |  |  |
| H:M ratio | 0.28 ± 0.17 | 0.29 ± 0.22 | 0.27 ± 0.15 | 0.20 ± 0.13 | 0.757 | 0.523 |
| MVIC torque, Nm/kg | 2.97 ± 0.48 | 3.50 ± 0.58[a] | 3.19 ± 0.44 | 3.39 ± 0.50[a] | 3.103 | **0.035** |
| CAR, % | 94.73 ± 7.25 | 95.32 ± 4.91 | 95.47 ± 3.87 | 93.51 ± 6.05 | 0.323 | 0.809 |
| LSI for MVIC torque, % | 83.82 ± 11.72 | – | 94.52 ± 10.27 | – | t = 2.514 | **0.019** |

Notes:
SD, standard deviation; ACLR, anterior cruciate ligament reconstruction group; H:M Ratio, hoffman reflex H-wave to M-wave ratio; MVIC, maximum voluntary isometric contraction (knee extension); CAR, central activation ratio; LSI, limb symmetry index (knee extension). Bold indicates a model with statistical significance.
[a] Indicates statistically different than ACLR involved in *post hoc* testing.

NRMSE than controls ($R^2 = 0.596$, $p = 0.045$, d = 0.39 [0.08–0.70]). With respect to mean torque output, the ACLR limb demonstrated lower torque during the 70% trials compared to the uninvolved limb ($R^2 = 0.856$, $p < 0.001$, d = 0.83 [0.40–1.26]) and contralateral control limb (d = 0.56 [0.12–1.00]), and during 100% trials compared to all other limbs ($R^2 = 0.872$, $p < 0.001$; uninvolved: d = 0.89 [0.47–1.31]; control: d = 0.59 [0.16–1.01]; control contralateral: d = 0.74 [0.30–1.17]).

A total of 14,491 MU (VM = 7,111: ACLR = 3,527, Control = 3,584; VL = 7,380: ACLR = 3,878, Control = 3,502) were included in these analyses. During 70% trials, the number of MUs identified in the VL was no different between groups or limbs ($R^2 = 0.554$, $p < 0.001$, no *post hoc* effects); however, there were fewer MUs identified in the VM of the ACLR group compared to controls ($R^2 = 0.501$, $p < 0.001$, d = 0.45 [0.15–0.75]). During 100% trials, the number of MUs identified in the involved ACLR limb was lower in the VM than both the control limb ($R^2 = 0.223$, $p = 0.008$, d = 0.56 [0.13–0.99]) and control

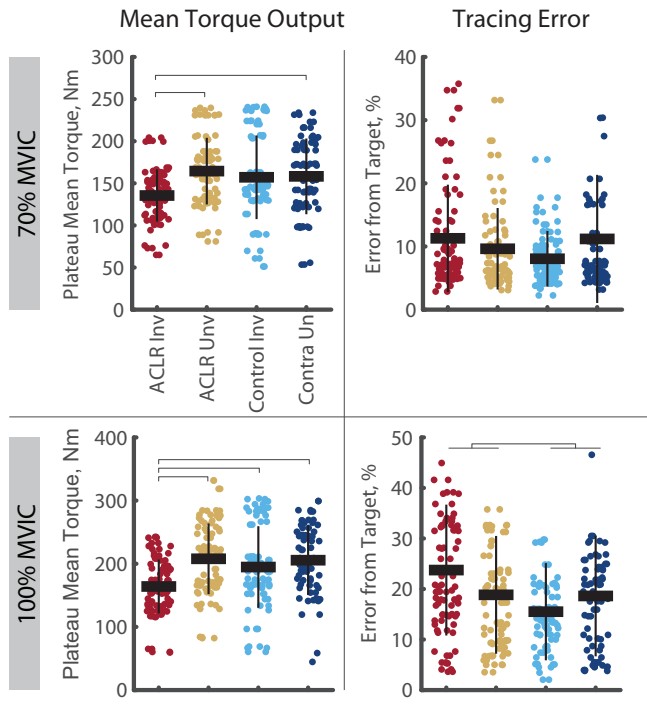

**Figure 2 Mean torque output and force tracing error during the 70% and 100% trials.** Tracing error is quantified using the normalized root mean square error. Individual circles depict the predicted value from the mixed effect model. Solid horizontal bars represent the least squared means and vertical error bars are the standard deviation. *Post hoc* comparisons that reached statistical significance are denoted with brackets. Group effects are denoted with the footed bracket. ACLR, anterior cruciate ligament reconstruction; MVIC, maximum voluntary isometric contraction; Nm, Newton-meter.

contralateral limb (d = 0.59 [0.14, 1.05]), and lower in the VL than both the uninvolved limb ($R^2$ = 0.297, $p$ = 0.002, d = 0.73 [0.31–1.15]) and control limbs (d = 0.76 [0.33–1.19]). The number of VM and VL MU for each group, limb and contraction intensity are shown in Fig. 3.

A significant group by limb difference was found for the mass-normalized RT range in both the VM and VL (Fig. 4). Here, the involved limb of the ACLR group demonstrated smaller RT range than the uninvolved side at both 70% (VM: $R^2$ = 0.640, $p$ = 0.025, d = 0.58 [0.15–1.01], VL: $R^2$ = 0.596, $p$ = 0.002, d = 0.75 [0.32–1.18]) and 100% trials (VM: $R^2$ = 0.683, $p$ = 0.007, d = 0.53 [0.11–0.95], VL: $R^2$ = 0.607, $p$ = 0.046, d = 0.76 [0.34–1.18]). Mass-normalized RT range of the VM was also smaller in the involved limb of the ACLR group compared to the control limb (d = 0.63 [0.20–1.06]) and control contralateral limb (d = 0.83 [0.36, 1.29]) during 100% trials. These differences were not observed with relative RT range (Fig. 5).

The analysis of mass-normalized MFR-RT slope revealed group by limb differences in both the VM and VL (Fig. 6). During the 70% trials, the ACLR involved limb demonstrated steeper slope than the uninvolved limb (VM: $R^2$ = 0.797, $p$ < 0.001, d = 0.85 [0.41–1.29]; VL: $R^2$ = 0.700, $p$ < 0.001, d = 1.19 [0.74–1.63]) and the control limb (VL: d = 1.03 [0.59–1.47]). During the 100% trials, the ACLR involved limb demonstrated steeper slope

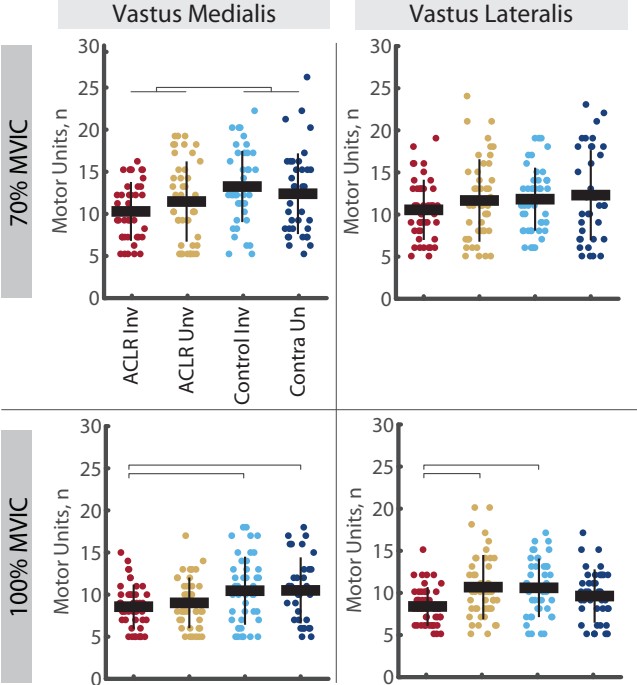

**Figure 3 Number of motor units identified in the vastus medialis and vastus lateralis during the 70% and 100% trials.** Individual circles depict the predicted value from the mixed effect model. Solid horizontal bars represent the least squared means and vertical error bars are the standard deviation. *Post hoc* comparisons that reached statistical significance are denoted with brackets. Group effects are denoted with the footed bracket. ACLR, anterior cruciate ligament reconstruction; MVIC, maximum voluntary isometric contraction; n, number (count).

than all other limbs; uninvolved (VM: $R^2 = 0.860$, $p = 0.009$, d = 0.66 [0.23–1.28]; VL: $R^2 = 0.889$, $p = 0.009$, d = 0.48 [0.07–0.89]), control (VM: d = 1.23 [0.77–1.69]; VL: d = 0.50 [0.07–0.93]), and control contralateral (VM: d = 0.95 [0.48–0.95]; VL: d = 0.27 [−0.14 to 0.69]). These differences in slopes were largely attenuated with relative RT (Fig. 7). During 70% trials, the ACLR involved limb retained steeper slope compared to the uninvolved limb (VL: $R^2 = 0.554$, $p < 0.001$, d = 0.41 [−0.02 to 0.83]) and control limb (d = 0.66 [0.23, 1.09]) during 70% trials, as well as compared to the control limb (VM: $R^2 = 0.853$, $p = 0.010$, d = 0.83 [0.37–1.29]) during 100% trials. Relative RT also revealed steeper slope in the control contralateral limb compared to the control limb during both 70% (VM: $R^2 = 0.710$, $p = 0.005$, d = 0.67 [0.22–1.11], VL: d = 0.77 [0.32–1.22]) and 100% trials (VL: VL: $R^2 = 0.802$, $p = 0.011$, d = 0.29 [−0.17 to 0.75]).

The analysis of mass normalized MUAP-RT slope revealed group by limb differences in the VM (Fig. 8). Here, the ACLR involved limb demonstrated a steeper slope than the uninvolved limb in the VM during the 70% ($R^2 = 0.879$, $p = 0.031$, d = 0.60 [0.17–1.03]) and 100% trials ($R^2 = 0.922$, $p = 0.001$, d = 0.59 [0.17–1.01]). There were no other group by limb differences in mass-normalized MUAP-RT slopes, nor were any differences observed in MUAP-RT slopes with relative RT (Fig. 9).

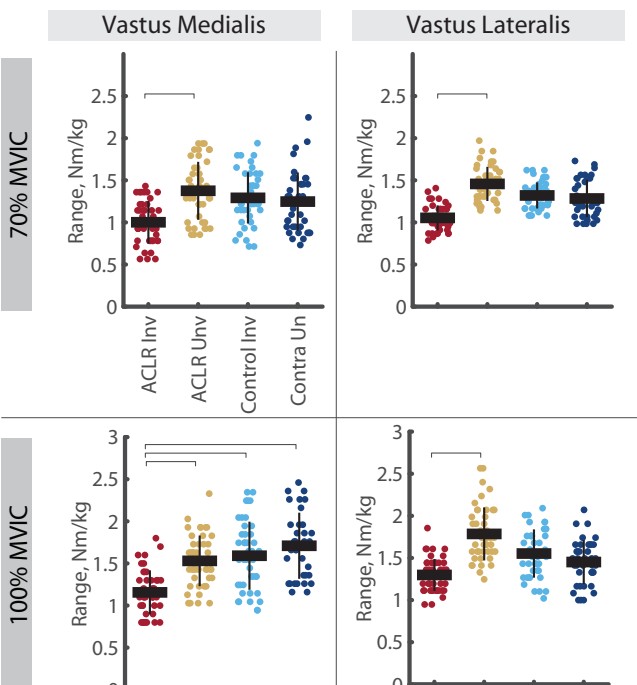

**Figure 4 Mass-normalized recruitment threshold range for MUs identified in the vastus medialis and vastus lateralis during the 70% and 100% trials.** Individual circles depict the predicted value from the mixed effect model. Solid horizontal bars represent the least squared means and vertical error bars are the standard deviation. *Post hoc* comparisons that reached statistical significance are denoted with brackets. ACLR, anterior cruciate ligament reconstruction; MVIC, maximum voluntary isometric contraction; Nm/kg, Newton-meter per kilogram.

## DISCUSSION

Our findings reveal differences in quadriceps MU properties in the involved limb of individuals with a history of ACLR during high intensity isometric contractions.

Our sample demonstrated large magnitude quadriceps weakness (d = 1.00) in the involved compared to uninvolved limb, which was accompanied by a smaller range of RTs, slower MU firing rates, and larger MU action potentials at high contraction intensities (70% and 100% MVIC). These differences were most pronounced when examining MU properties with mass-normalized RT and were attenuated when using relative RT, suggesting these impairments may underlie the loss of force generating capacity after ACLR and, to a lesser extent, changes in the MU recruitment schema as it relates intrinsically when normalized to individuals' MVIC. Our findings conflict with our hypothesis, which was consistent with prevailing theory that inhibition of the motor neuron pool following ACLR would necessitate upregulation of MU firing rates to compensate for smaller MU size. Conversely, our findings suggest a limited ability to recruit MUs and to upregulate the firing rates of recruited MUs, resulting in fewer active MUs that fire more slowly at given force outputs. Key features of our sample, including full voluntary quadriceps activation and normal spinal reflexive excitability, support our interpretation of impairments in both motor

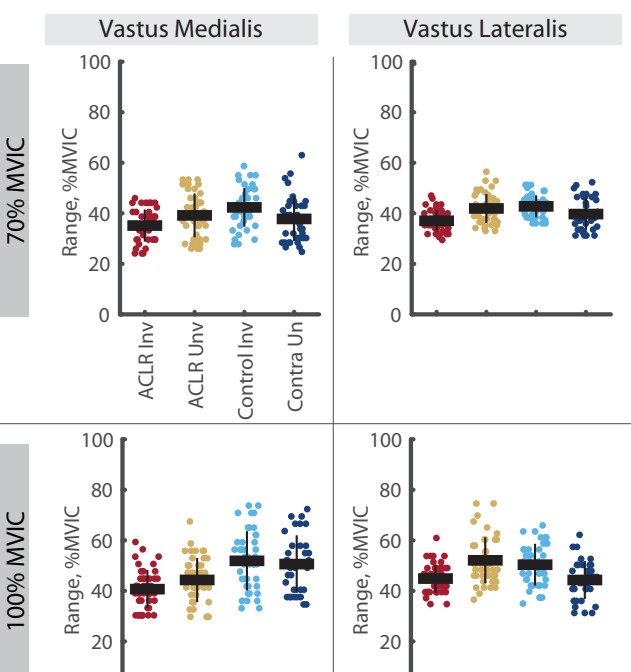

**Figure 5 Relative recruitment threshold range for MUs identified in the vastus medialis and vastus lateralis during the 70% and 100% trials.** Individual circles depict the predicted value from the mixed effect model. Solid horizontal bars represent the least squared means and vertical error bars are the standard deviation. *Post hoc* comparisons that reached statistical significance are denoted with brackets. ACLR, anterior cruciate ligament reconstruction; MVIC, maximum voluntary isometric contraction.

neuron and morphological properties of muscle fibers. Accordingly, we discuss these results in the context of previously theorized reductions in cortical neural drive to the quadriceps (*Nuccio et al., 2021*; *Sherman et al., 2023*), muscle fiber type transitions (*Lepley et al., 2020*; *Noehren et al., 2016*) and reductions in quadriceps functional capacity due to catabolism of high threshold MUs following ACL injury or surgery (*Hunt et al., 2022*).

Shorter mass-normalized RT range and slower MU firing rates at given RTs suggests an inability to upregulate MU firing rates may account for reduced quadriceps force generating capacity. Recently, a similar cross-sectional analysis (*Nuccio et al., 2021*) also reported an inability to upregulate MFR in the ACLR limb compared contralaterally in a small sample of individuals approximately 8 months post-operative from ACLR. The authors' attributed this finding to decreased excitatory input to the motor neuron pool. In healthy muscle, MU firing rates are determined by the net synaptic input received by motoneurons and the intrinsic properties of these neurons (*Heckman & Enoka, 2012*). For instance, inherent factors like persistent inward currents (PICs) can significantly enhance motoneuron excitability and, consequently, MU firing rates and rate coding after initial activation (*Heckman & Enoka, 2012*; *Heckman et al., 2008*). On the other hand, increased inhibitory inputs *via* reflexive inhibition, can depress PICs, leading to reduced ability to upregulate MU firing rates (*Hyngstrom et al., 2007*; *Revill & Fuglevand, 2017*).

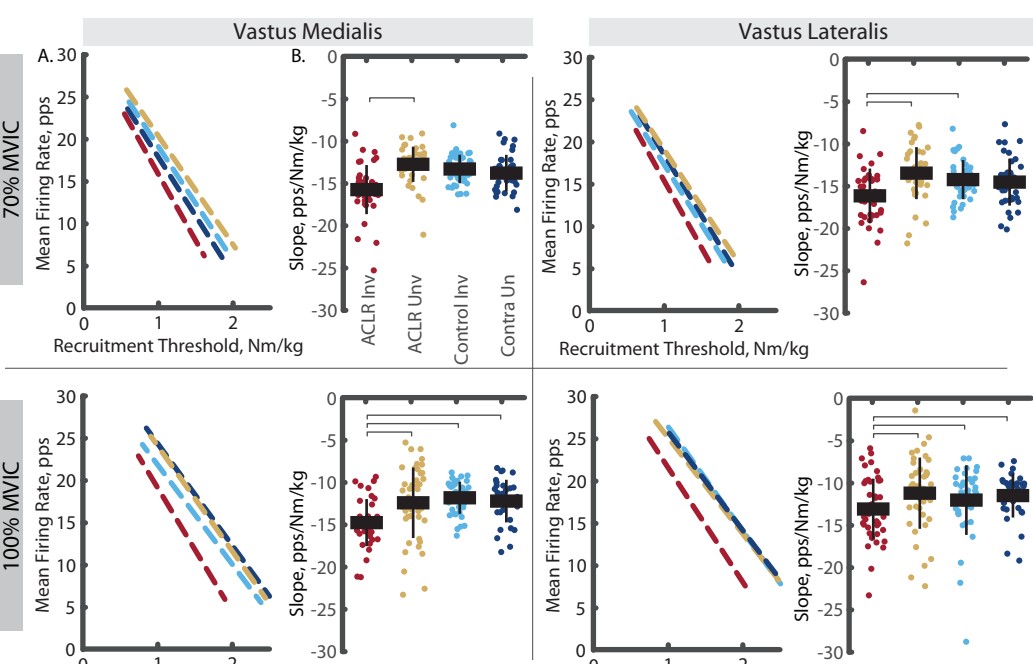

**Figure 6 Mean firing rate to mass-normalized recruitment threshold slope comparisons for MUs identified in the vastus medialis and vastus lateralis during the 70% and 100% trials.** (A) Dashed lines represent the mean firing rate to recruitment threshold relationship across the group*limb levels. Lines are plotted with average slopes and mass-normalized recruitment threshold ranges without controlling for y-intercept. (B) Individual circles depict the predicted value from the mixed effect model. Solid horizontal bars represent the least squared means and vertical error bars are the standard deviation. *Post hoc* comparisons that reached statistical significance are denoted with brackets. ACLR, anterior cruciate ligament reconstruction; MVIC, maximum voluntary isometric contraction; Nm/kg, Newton-meter per kilogram; pps, pulses per second.

By recording normal motor neuron pool excitability and volitional activation in our sample, we overcome key limitations in the interpretation of diminished excitatory input with respect to volitional capacity and conclude that post-synaptic inhibition was not influencing MU behavior in this sample. Yet, we identified insufficient recovery of strength, failure to recruit MUs through a large RT range, and inability to upregulate firing rates of recruited MUs during high-volitional efforts compared to contralateral and control limbs. Accordingly, we speculate that the neural deficit of the involved limb in this sample is the result of reduced cortical drive, rather than alterations in spinal reflexive circuits. Our sample was 2.4 years from surgery on average, a time point when spinal inhibition caused by arthrogenous pain and effusion is largely resolved (*Lepley et al., 2015*; *Rush, Glaviano & Norte, 2021*). Conversely, reduced corticospinal excitability and cortical drive develop during periods of rehabilitation (*Rush, Glaviano & Norte, 2021*; *Sherman et al., 2023*), persist into the long term, and contribute to persistent structural atrophy and reduced muscle capacity (*Birchmeier et al., 2020*; *Lisee et al., 2019*).

The description of motor neuron pool excitability (as quantified by the Hoffmann Reflex) and voluntary muscle activation (as quantified by the central activation ratio) are

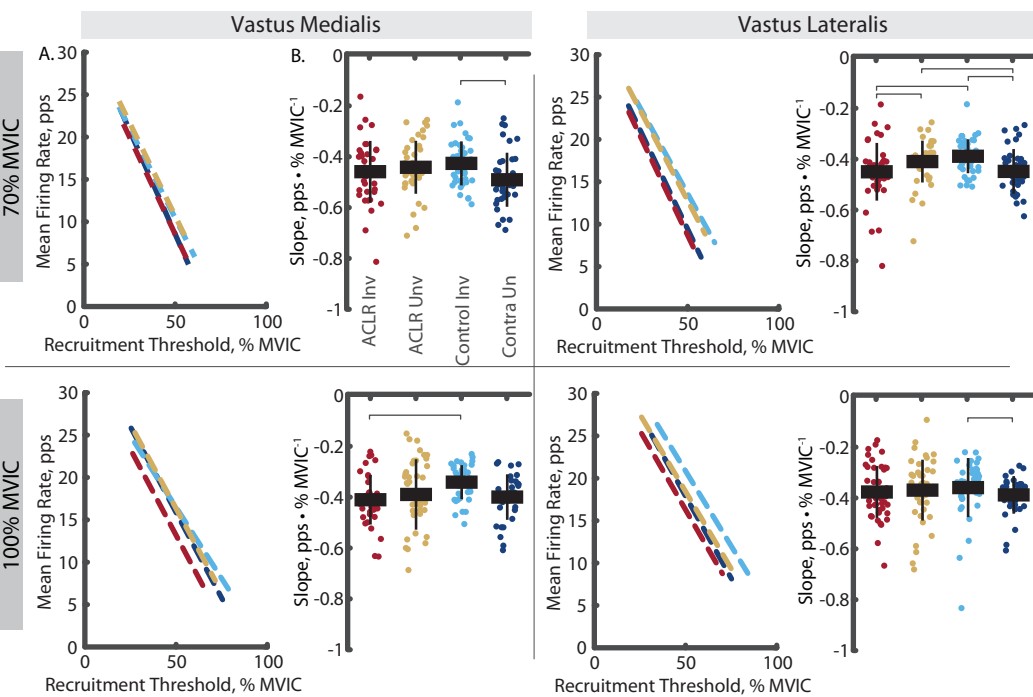

**Figure 7 Mean firing rate to relative recruitment threshold slope comparisons for MUs identified in the vastus medialis and vastus lateralis during the 70% and 100% trials.** (A) Dashed lines represent the mean firing rate to recruitment threshold relationship across the group*limb levels. Lines are plotted with average slopes and relative recruitment threshold ranges without controlling for y-intercept. (B) Individual circles depict the predicted value from the mixed effect model. Solid horizontal bars represent the least squared means and vertical error bars are the standard deviation. *Post hoc* comparisons that reached statistical significance are denoted with brackets. ACLR, anterior cruciate ligament reconstruction; MVIC, maximum voluntary isometric contraction; Nm/kg, Newton-meter per kilogram; pps, pulses per second.

key strengths of this study. The absence of impairment in motor neuron pool excitability and voluntary activation in our sample indicates all groups were able to recruit a normal proportion of their motor neuron pool. Considered in the presence of muscle weakness and adjusting for decomposition quality (*i.e.*, number of MUs), shorter RT range in the ACLR limb represents less robust recruitment of MUs at high RTs (Fig. 8). In other words, rather than an inability to activate all MUs in the motor neuron pool, our findings may suggest a catabolism of motor units has resulted in a smaller motor neuron pool and a recruited MU distribution that is less dense than the other limbs. This interpretation can explain the ability to recruit a normal proportion of the motor neuron pool despite presenting with reduced strength and altered mass-normalized RT range, MU firing rates, and action potential sizes. When normalized to MVIC, the ACLR limb recruitment schema appears normal. However, this can be misleading as MVIC normalization obscures differences in absolute strength between groups and limbs, and thereby contributes to artificially flattened slopes (cf. Figs. 6 and 7). Conversely, when normalized extrinsically to force output (Nm/kg), the ACLR limb recruitment schema shows slower firing rates (in both VM and VL) and larger amplitudes (in VM) at lower recruitment threshold than the

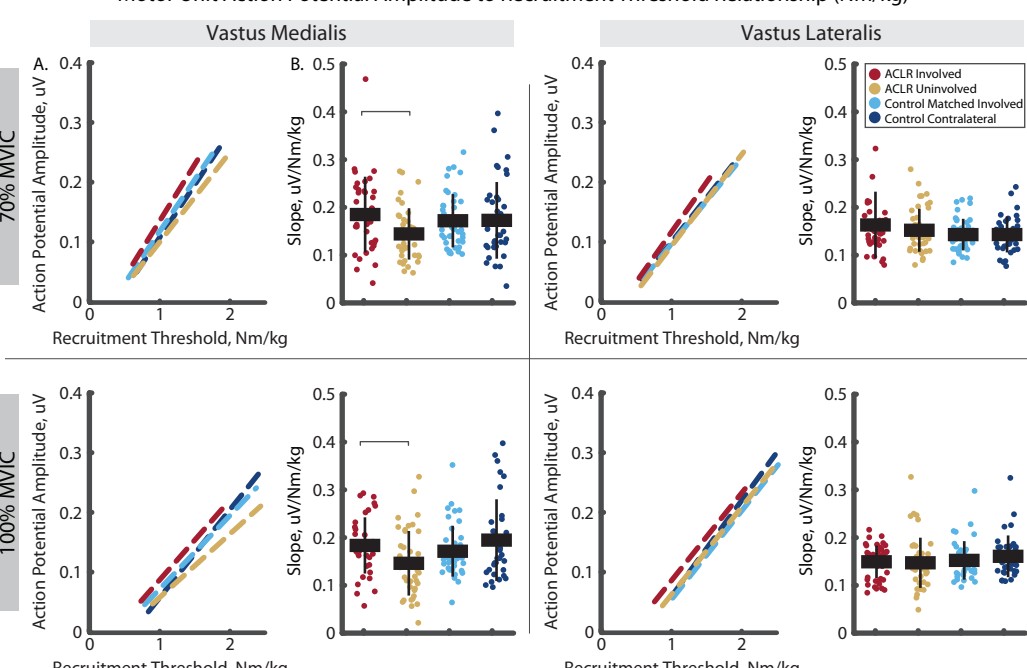

**Figure 8** Motor unit action potential amplitude to mass-normalized recruitment threshold slope comparisons for MUs identified in the vastus medialis and vastus lateralis during the 70% and 100% trials. (A) Dashed lines represent the action potential amplitude to recruitment threshold relationship across the group*limb levels. Lines are plotted with average slopes and mass–normalized recruitment threshold ranges without controlling for y-intercept. (B) Individual circles depict the predicted value from the mixed effect model. Solid horizontal bars represent the least squared means and vertical error bars are the standard deviation. *Post hoc* comparisons that reached statistical significance are denoted with brackets. ACLR, anterior cruciate ligament reconstruction; MVIC, maximum voluntary isometric contraction; Nm/kg, Newton-meter per kilogram; uV, microvolts.

contralateral limb, suggesting that a smaller motor neuron pool necessities recruitment of available motor units at lower force output. Multiple studies (*Lepley et al., 2015*; *Otzel, Chow & Tillman, 2015*; *Pietrosimone et al., 2015*) report similar quadriceps voluntary activation (>90% CAR in ACLR involved limb) despite quadriceps muscle weakness at similar times from surgery (>25 months). Motor neuron pool excitability shows a similar time course of resolution by 2 years following surgery (*Rush, Glaviano & Norte, 2021*). As ratio measures, CAR and H-Reflex techniques may be too crude for adequate description of nuanced MU behavior this far removed from surgery and highlights the importance of new tools, such as decomposition EMG in the context of neurophysiological activation and motor output. Although motor neuron pool size is not directly measurable, we believe our interpretation of these results is supported in the literature examining the neurometabolic cascade and quadriceps muscle fiber type transitions stemming from ACL injury.

A MU consists of an alpha motor neuron and all the muscles fibers it innervates. In healthy muscle tissue, MU are homogenous of type I or type II muscle fibers (*Noehren et al., 2016*; *Pette & Staron, 2000*). Following ACL injury, altered neural activity during

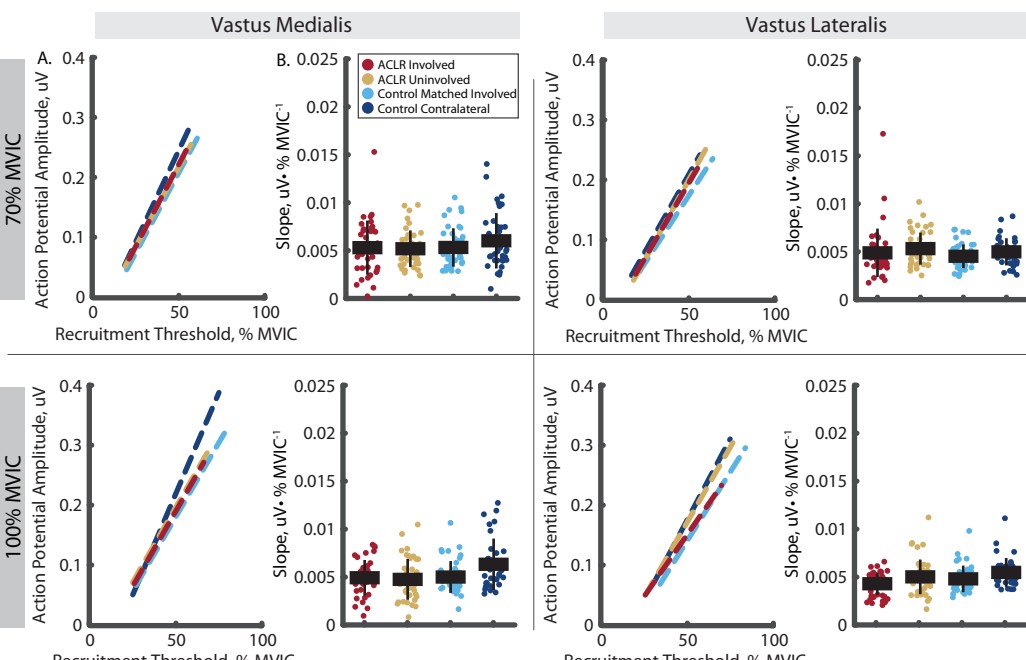

**Figure 9 Motor unit action potential amplitude to relative recruitment threshold slope comparisons for MUs identified in the vastus medialis and vastus lateralis during the 70% and 100% trials.** (A) Dashed lines represent the action potential amplitude to recruitment threshold relationship across the group*limb levels. Lines are plotted with average slopes and relative recruitment threshold ranges without controlling for y-intercept. (B) Individual circles depict the predicted value from the mixed effect model. Solid horizontal bars represent the least squared means and vertical error bars are the standard deviation. *Post hoc* comparisons that reached statistical significance are denoted with brackets. ACLR, anterior cruciate ligament reconstruction; MVIC, maximum voluntary isometric contraction; Nm/kg, Newton-meter per kilogram; PPS, pulses per second.   

periods of arthrogenic muscle inhibition likely leads to progressive reductions in neuromuscular junctions and motoneurons in the periphery resulting in the accumulation of denervated fibers within the involved quadriceps muscle (*Hunt et al., 2022*; *Lepley et al., 2020*). Theoretically, the change in afferent signaling, reduced excitability of the motor neuron pool, and inability to recruit MU severely limit the ability to engage the muscle during exercise. In rat models, these neuromuscular changes precede atrophic factors in the quadriceps (*Hunt et al., 2022*), suggesting anabolism would occur only within MUs activated during rehabilitation, while catabolism of non-activated (inhibited) MUs may occur with exposure to catabolic substances, such as inflammatory cytokines. This may result in successful reconditioning and habituation of small amplitude, low-threshold MUs of predominantly type I fibers, while larger amplitude, high-threshold MUs of predominantly type IIa fibers remain untrained despite rehabilitation. This strategy of predominant recruitment of low-threshold MU is theoretically sufficient during periods of low volitional effort (<2 Nm/kg). However, during periods of high volitional effort (>2 Nm/kg), these results suggest an inability to sufficiently recruit large amplitude high-threshold MUs. Additionally, over time, this denervation-reinnervation cycle can lead to transitions of muscle phenotypes, resulting in more MU with heterogeneous fiber

types or slow twitch- to -fast-twitch fiber type transitions (*Pette & Staron, 2000*). Such transitions are observed within the VL during post-operative periods following ACLR and suggest that early motor neuron pool inhibition may play a role in subsequent MU and fiber type deconditioning.

One alternative interpretation of these results might be that muscle fiber atrophy would explain the lesser force generating capacity of recruited MUs. Alas, we did not measure muscle morphology, so we cannot entirely rule this out. However, MU action potential size is related to muscle fiber size (*Hakansson, 1956*) and the number of innervated muscle fibers (*Milner-Brown, Stein & Yemm, 1973*). Furthermore, MU action potential size increases at the same relative RTs when muscle CSA is increased with resistance training (*Pope et al., 2016*), and decreases in relation to reductions in muscle size during immobilization (*Inns et al., 2022*). Thus, atrophy of higher-threshold MUs after ACLR would lead to both shorter RT range and lower $MUAP_{AMP}$-RT slope. Our results demonstrate shorter RT range with increased or unchanged $MUAP_{AMP}$-RT slope, suggesting atrophy of muscle fibers is a less suitable explanation.

These findings extend the work of *Nuccio et al. (2021)* even when considered in context of experimental differences. Differences between this previous work and ours include variations in analysis (*e.g.*, different dependent variables, pooling of contraction intensities, statistical methods, and use of $MUAP_{AMP}$). Most notably, the firing rate dysregulation reported by *Nuccio et al. (2021)* was only present when normalizing RT to MVIC and not when normalizing RT to absolute torque. Mass-normalized torque is used to characterize sufficiency of quadriceps strength (>3.0 Nm/kg) in active populations and is a standard of clinical recovery after ACLR (*Kuenze et al., 2015*). Our inclusion of mass-normalized RT, in addition to relative RT, helps to contextualize the MU properties underlying clinical muscle weakness after ACLR.

Our results must be weighed against three pertinent limitations. Namely, examination of MU properties is flawed as it relies heavily on the "blind" algorithmic identification of MU from gross EMG signals sampling only a small portion of each muscle. As such, defining RT range by 95% confidence bounds is highly dependent upon identified motor units despite the likelihood that unidentified motor units were active outside those bounds. Despite this, we feel RT range is a worthwhile outcome despite these limitations given there is no reason to suspect differences in decomposition quality between limbs or groups, especially in light similar quadriceps volitional activation. The additional methodological steps (*Del Vecchio et al., 2020*) and the application of a mixed effect model that accounted for within and between participant variability and differences in decomposition quality help to further address this limitation and support our interpretations regarding the drivers of persistent quadriceps weakness after ACLR. Second, a consideration when designing this study was whether the procedures would induce fatigue, thereby affecting MU properties in a manner not intended (*Stock & Mota, 2017*). To address this, participants had 10 min of rest between MVIC and CAR testing prior to starting the experimental trapezoid contractions. Further, we randomized trapezoidal contraction intensities and ensured adequate rest between trials (≥1 min). Lastly, our sample of participants is a wide range of time from surgery (7 to 79 months). We cannot determine

whether time from surgery may have influenced our findings, although exploratory correlations found no associations between time from surgery and our dependent variables. Combined with the sample's high levels of physical activity, history of resistance training, and return to unrestricted physical activity, this supports our interpretation of chronic neural impairments underlying persistent quadriceps weakness.

## CONCLUSION

In conclusion, our study highlights differences in quadriceps MU properties in individuals with a history of ACLR. The findings suggest limited ability to recruit MUs and upregulate firing rates of recruited MUs, which may contribute to the observed quadriceps weakness and reduced force-generating capacity. These findings support theories of reduced cortical excitability and descending cortical drive influencing MU properties and resultant quadriceps muscle function in a sample with chronic ACLR. Further research is needed to elucidate the underlying mechanisms driving these differences in MU properties and their functional implications.

### Funding

This work was funded by the NATA Research & Education Foundation (1819DGP05) and the University of Toledo deArce-Koch Memorial Endowment Fund. The funders had no role in study design, data collection and analysis, decision to publish, or preparation of the manuscript.

### Grant Disclosures

The following grant information was disclosed by the authors:
NATA Research & Education Foundation: 1819DGP05.
University of Toledo deArce-Koch Memorial Endowment Fund.

### Competing Interests

David A. Sherman is an owner of Live4 Physical Therapy and Wellness. The authors declare that they have no other competing interests.

### Author Contributions

- David A. Sherman conceived and designed the experiments, performed the experiments, analyzed the data, prepared figures and/or tables, authored or reviewed drafts of the article, and approved the final draft.
- Justin Rush conceived and designed the experiments, performed the experiments, authored or reviewed drafts of the article, and approved the final draft.
- Matt S. Stock conceived and designed the experiments, analyzed the data, authored or reviewed drafts of the article, and approved the final draft.
- Christopher D. Ingersoll conceived and designed the experiments, authored or reviewed drafts of the article, and approved the final draft.

- Grant E. Norte conceived and designed the experiments, analyzed the data, prepared figures and/or tables, authored or reviewed drafts of the article, and approved the final draft.

## Human Ethics

The following information was supplied relating to ethical approvals (*i.e.*, approving body and any reference numbers):

University of Toledo granted Ethical approval to carry out the study within its facilities.

## Data Availability

The data is available at Open Science Framework: Sherman, David. 2023. "Neural Drive and Motor Unit Characteristics after Anterior Cruciate Ligament Reconstruction: Implications for Quadriceps Weakness." OSF. July 28. DOI 10.17605/OSF.IO/6DC3X.

## Supplemental Information

Supplemental information for this article can be found online at http://dx.doi.org/10.7717/peerj.16261#supplemental-information.

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
