# Peer review of "Neural drive and motor unit characteristics after anterior cruciate ligament reconstruction: implications for quadriceps weakness"

_PeerJ, doi:10.7717/peerj.16261_

## Round 0.1 · original submission · Major Revisions

Please be sure to address the methodological concerns that were brought up by Reviewers 1 and 3. Specifically, explain and justify the unique approach (compared to previous research) taken to assess motor units and answer your research questions.

Reviewer 1 ·

Basic reporting

Overall
This is interesting work by the authors attempting differences in neural control knee muscles post ACL surgery compared to the uninvolved leg and a control group. The author are commended on their robust design and comprehensive assessments to make neuromuscular inferences. The manuscript is well written with a strong rationale that supports the importance of this work. However, I have a few concerns that may affect the inferences being made by the authors. Namely, I have concerns and questions about the small MU yield and decision to normalize recruitment threshold to body mass, despite a strength difference in the involved limb. Also, there are analysis criteria typical of papers using MU decomposition not used in this study.

Experimental design

The authors are commended for achieving the sample size they did given the challenges of recruiting this specific population. However, the variability in time from surgery is large. The SD is quite large, so it would appear there were individuals in the ACLR group who were 7 months out from their surgery and others who were several years. It would seem likely that this discrepancy would affect outcomes in this group.

Familiarization is critical to reliable testing of voluntary activation and neuromuscular testing involving novel tasks like tracings. Why was a familiarization visit not performed, and how did it not affect your outcomes?

Participants performed a total of 18 knee extension contractions, that’s a lot. Particularly since MU data was derived from tracings after 15 contractions (3 for VA and 15 practice trials), how do we know fatigue didn’t affect the data? It is quite possible that those in the ACLR would be more likely to accumulate fatigue even with randomization of tracing intensities.

It is unclear why absolute RT is normalized to mass. Why did you not use relative RT (%MVIC)? Since strength is reduced for the involved leg, to appropriately make inferences about MU size, RT needs to be relative to RT. Also, it is not common to normalize to mass so reduces comparability. Also, the subjects were similar in stature. Finally, this assumes greater torque accompanies greater body mass, which is not verified in this sample. In other words, there is no reason for RT to be normalized to mass, and this greatly reduces comparability to MU literature and may be inappropriate for the inferences being made.

It would be important to determine if the results differ when RT is relative to %MVIC.
Was the 100% MVIC tracing a true MVIC? In other words, how did the peak torque differ from the MVIC performed earlier in the visit? Was this difference different for the involved leg vs. uninvolved leg or control group?

Validity of the findings

The number of MUs identified for analysis are very low compared to similar literature using the 5-pin decomposition. Most studies analyze greater than 15 MUs using the same criteria (Herda et al. 2021, Colquhoun et al. 2018, Pope et al. 2016, Jenkins et al., 2021) . Why was your yield so low? Can relationships be relied up with fewer than 10 or 12 data points? Also, based on your average, it would seem like some contractions had 4 or less MUs for a yield. Most other work referenced above has required at least 5 MUs for a contraction to be used for analysis. In addition, other criteria such as at least a 20% RT range, RT that is within 20% of the target contraction intensity.


At least this significant relationships mentioned at the end of the results should be displayed since they provide further support for primary findings/hypotheses.
Discussion

The ACLR group could have demonstrated greater instability about the joint and thus greater torque fluctuations during tracings. Reporting the CV for torque during the plateau phase would help strength this cross-sectional study.

·

Basic reporting

Thank you for the opportunity to review this paper. I commend the authors for this work, identifying a gap in knowledge of the neuromuscular mechanism of persistent quadriceps weakness after ACL reconstruction, utilizing appropriate instrumentation and methodology to collect data, and analyzing and interpreting the data in a meaningful way.

The introduction starts out clearly (lines 49-82), providing context and setting up for the rationale of the study. However, from line 83 to 129, the use of jargon without adequate explanation makes it difficult to follow. For example, lines 83-92 mention two dependent variables that are used in this study (i.e., MUAP_AMP and MFR-RT). It is unclear from reading the text what these variables mean and what an increase or decrease in MUAP_AMP-RT and MFR-RT slope and length mean. Rather than explaining in terms of dependent variables (i.e., MUAP_AMP and MFR-RT), I recommend that the authors discuss the constructs or phenomena in the physiological context (e.g., MU atrophy, inability to recruit MU, muscle fiber type changes). The authors actually do this in the discussion (lines 331-383) - e.g., "Here, full MU recruitment (i.e., high CAR), no difference in MUAP size relative to recruitment threshold (i.e., MUAP_AMP-RT slope), and shorter MUAP_AMP-RT recruitment line length, suggest these MUs are incapable of recruitment and not simply atrophied." (lines 353-6) Moving parts of this section, which is well-written, to replace parts of the introduction filled with technical language may improve clarity. I would also encourage the authors to consider revising their hypothesis to one without use of the dependent variables.

The discussion states "...large magnitude quadriceps weakness" (line 318) - what is the magnitude based on? Statistical effect size or some measure of clinically important difference? Later in the discussion states "... insufficient quadriceps strength (< 3.0 Nm/kg..." (line 417) - please provide a reference for where this threshold was taken from.

Table 1 - please provide the range for "Time after Surgery." Looking at the SD, I suspect that there is quite a bit of variability in the time since surgery and this provides context for interpreting the study results.

Tables 3 and 4 - the numbers 30, 50, 70, and 100 need to be labeled.

Figure 1 - what do "Acc 94.3, Acc 93.0..." mean?

Figures 2 and 3 - does slope and line segment length have any unit of measurement?

Experimental design

Participants performed "trapezoid" contractions at 30, 50, 70, and 100% MVIC. The term trapezoid contraction should be used after explaining what this means. Also, rationale for using a wide range of contraction intensities is unclear. Wouldn't the effect of prior ACL reconstruction on MU recruitment and firing rates be relevant at the higher intensities? With the 4 levels of contraction intensity, measured on both lower limbs in both groups, 5 variables (# MUs, MFR-RT slope, MFR-RT length, MUAP_AMP-RT slope, MUAP_AMP-RT length) derived from essentially the same EMG data, and separate inferences made for these scenarios, I would be concerned about increased risk of type I error.

Please provide a rationale for selecting 90 degrees of knee flexion as the position of testing (lines 173-4).

Validity of the findings

The findings are reported clearly based on appropriate data analysis.

Again, because the introduction was written with too much technical language, the original research question was not clear. The discussion does a much better job at this so I would encourage using similar type of language to revise the introduction.

·

Basic reporting

The authors have conducted an interesting study that could potentially have an impact in this field of study. My major concerns are with the methods used to analyze the motor unit relationships. It is difficult to interpret the data since recruitment thresholds were not quantified in a similar manner to other research in this field of study. Also, it is unclear the physiological meaningfulness of the “length” variable. Most of the physiological interpretations are based on the “length” variable. I would also encourage the authors to remove the 30% and 50% MVCs for simplicity since they are interested in higher-threshold motor units.
Line 114: Further explanation is needed regarding the meaningfulness of the length of the regression line. Similar slopes of the MUAP-RT relationships would suggest there would be similar activation levels regardless of the length of line, correct? The “end of the line” doesn’t necessarily mean there were no motor units recruited beyond that point. Similar to the “start of the regression line” as there were motor units recruited prior to the start of the regression line. The line is determined by the recorded motor units. The length of the line could simply be a function of recorded recruitment threshold ranges.
The authors are encouraged to provide a figure to illustrate the methods used for quantifying the motor unit relationships - including the length variable. This could be done for data presented in Figure 1. Would the length of the line reflect the recorded recruitment threshold range?
It is unclear why the recruitment threshold was not normalized to peak torque (Nm) during the MIVC. What is the rationale for normalizing recruitment threshold to mass? The authors should provide a rationale for this procedure. Normalizing recruitment threshold to isometric peak torque is common and can be easily compared to other literature.
Line 234: The authors should provide a rationale for the covariates. MVIC wouldn’t need to be a covariate if peak torque was used for normalizing recruitment threshold, correct? Why did the authors use H-reflex and CAR as covariates. Was h-reflex from the VL used as a covariate for VM motor unit data? According to the results, CAR and H-reflex were similar between groups so I am not sure why it would be a covariate. Nonetheless, the authors should provide a rationale for the chosen covariates. I would recommend not using those variables as covariates.
Line 263: The authors should report the recruitment threshold ranges for contraction intensities and muscles. Are the ranges similar between groups and limbs for each contraction? The authors should report the strength of the relationships also, this is particularly true since recruitment thresholds were not normalized to peak torque.
The results section is complete and thorough. However, a strong rationale needs to be presented regarding the normalizing procedures and covariates. I would encourage the authors to normalize motor unit parameters to MVIC rather than including body mass. Also, do the covariates change the outcomes of the stat tests?
Figure 1. Did the authors really average the motor unit amplitude from the 5 second period of the plateaued region? Or was the amplitude averaged across the entire action potential train. It looks like the methods indicates it was averaged across the 4 channels using the entire action potential chains.

Experimental design

The design is acceptable.

Validity of the findings

Authors should consider more traditional methods to quantify the motor unit relationships.

---

## Round 0.2 · Minor Revisions

Please address Reviewer #1's concerns about your Discussion and Conclusion.

Reviewer 1 ·

Basic reporting

no comment

Experimental design

no comment

Validity of the findings

The degree of variability for tracing error and the relatively greater mean in the involved limb appears at least relatively robust. The involved limb appears to be driving the group difference for tracing error. Since tracing fidelity is critical to identification of MUs, how do we know these relative differences in tracing error are influential in the RT range?

Discussion; Line 413: It is very interesting that the results were dependent upon the normalization method. However, it is unclear what is meant by “might be partially explained by MU properties”. A MU consists of the neuron, its axon, and the innervated fibers. Thus, this could be referring to intrinsic neuronal properties or morphological properties of muscle fibers. Albeit this is the first paragraph of the Discussion, the concept should be made clearer. What do the divergent findings mean?

As stated in the Introduction, an inability to voluntarily activate the muscle (CAR) is typical in ACLR groups. Why was this not present here, despite the findings indicated reduced neural drive? This should be discussed.

Discussion; Lines 485 – 495: This argument is flawed. CAR does not assess the ability to recruit a normal number of MUs, but rather all of the MUs. It is a standard assessment to determine if there is a neural deficit, regardless of location. If subjects were unable to recruit additional, higher threshold MUs (as suggested by interpretation of MU findings), why was a greater superimposed torque not induced during this assessment leading to a diminished CAR. Can it be explained why there were differences in MU findings between legs, which accompanied a strength difference, but this did not show up in CAR?

Discussion, Lines 502 – 506: Similarly, full voluntary activation would not mean that higher threshold MUs are incapable of recruited. Indeed, incomplete voluntary activation is commonly interpreted as the opposite in the context of aging or disuse.

Discussion, Lines 508 – 510: This is an overzealous interpretation of your self-reported, descriptive data, and this statement is unrelated to the purpose of this study. These data suggest some were collegiate athletes whereas some were not, and some performed resistance training and others not. Regardless, there was no intervention and given the nature of this self-reported data, it is overstepping to provide even speculative conclusions regarding these data.

Conclusions, Lines 595 – 597: Here and in the other locations where it is mentioned there is an inability to recruit “new” MUs. Rephrasing is recommended. New suggests the development of new neuron, etc.. If reinnervation is being considered, again, “new” is not the best word. Also, we don’t know for sure that there was death of MUs in your sample. I think you are referring to the inability to recruit higher threshold MUs.

Additional comments

Figure 8 has an incorrect title as it refers to firing rates but depicts action potential size.

·

Basic reporting

The authors have adequately addressed the issues of clarity I raised in the first review.

Experimental design

The authors have clarified the methodological questions I had in the first review.

Validity of the findings

No comment.

---

## Round 0.3 · accepted · Accept

Thank you for responding to the reviewers' comments. I have reviewed the current version and believe that it is ready for publication.